# Sustained Release of Linezolid from Prepared Hydrogels with Polyvinyl Alcohol and Aliphatic Dicarboxylic Acids of Variable Chain Lengths

**DOI:** 10.3390/pharmaceutics12100982

**Published:** 2020-10-17

**Authors:** Gustavo Carreño, Adolfo Marican, Sekar Vijayakumar, Oscar Valdés, Gustavo Cabrera-Barjas, Johanna Castaño, Esteban F. Durán-Lara

**Affiliations:** 1Instituto de Química de Recursos Naturales, Universidad de Talca, Talca 3460000, Maule, Chile; gcarreno@utalca.cl (G.C.); amarican@utalca.cl (A.M.); 2Bio and NanoMaterials Lab, Drug Delivery and Controlled Release, Universidad de Talca, Talca 3460000, Maule, Chile; 3Marine College, Shandong University, Weihai 264209, China; vijaysekar05@gmail.com; 4Vicerrectoría de Investigación y Postgrado, Universidad Católica del Maule, Talca 3460000, Maule, Chile; ovaldes@ucm.cl; 5Unidad de Desarrollo Tecnológico, Universidad de Concepción, Av. Cordillera 2634, Parque Industrial Coronel, Coronel 4191996, Biobío, Chile; g.cabrera@udt.cl; 6Facultad de Ingeniería y Tecnología, Universidad San Sebastián, Lientur 1457, Concepción 4080871, Chile; johanna.casta@gmail.com; 7Departamento de Microbiología, Facultad de Ciencias de la Salud, Universidad de Talca, Talca 3460000, Maule, Chile

**Keywords:** hydrogel, polyvinyl alcohol, aliphatic dicarboxylic acids, sustained release, linezolid, equilibrium swelling ratio, accumulative release, thermogravimetric analysis

## Abstract

A series of hydrogels with a specific release profile of linezolid was successfully synthesized. The hydrogels were synthesized by cross-linking polyvinyl alcohol (PVA) and aliphatic dicarboxylic acids, which include succinic acid (SA), glutaric acid (GA), and adipic acid (AA). The three crosslinked hydrogels were prepared by esterification and characterized by equilibrium swelling ratio, infrared spectroscopy, thermogravimetric analysis, mechanical properties, and scanning electron microscopy. The release kinetics studies of the linezolid from prepared hydrogels were investigated by cumulative drug release and quantified by chromatographic techniques. Mathematical models were carried out to understand the behavior of the linezolid release. These data revealed that the sustained release of linezolid depends on the aliphatic dicarboxylic acid chain length, their polarity, as well as the hydrogel crosslinking degree and mechanical properties. The in vitro antibacterial assay of hydrogel formulations was assessed in an *Enterococcus faecium* bacterial strain, showing a significant activity over time. The antibacterial results were consistent with cumulative release assays. Thus, these results demonstrated that the aliphatic dicarboxylic acids used as crosslinkers in the PVA hydrogels were a determining factor in the antibiotic release profile.

## 1. Introduction

Multidrug-resistant (MDR) bacteria or “superbugs” represent one of the most important challenges to public health and pose a huge economic burden on global health care [1,2]. Indeed, antibacterial resistance causes 700,000 deaths per year worldwide [3]. The World Health Organization (WHO) has classified *Enterococcus faecium* as one of the primary drug-resistant pathogens posing the most significant risk to public health. This bacterium causes urinary tract infections, hospital-acquired bloodstream infections, abdominal and pelvic abscesses, endocarditis, and chronic periodontitis. The importance of this pathogen in these types of infection is reinforced by their intrinsic and acquired resistance to various antimicrobial agents, which renders them challenging to treat [4]. Linezolid is among the few available antibiotics that can treat bacterial resistance. This antibiotic is the first clinically useful oxazolidinone antibacterial agent [5]. This chemotherapeutic agent has been approved for the treatment of complicated skin and skin-structure tissue infections principally caused by vancomycin-resistant *E. faecium* [6]. Recent studies indicate that the effectiveness of antibacterial agents is better when they are released through drug delivery systems. Furthermore, these systems could slow down the progression of bacterial resistance to antibiotics [7,8]. In this context, hydrogels appear to be excellent candidates as antibiotic delivery platforms [9]. Hydrogels are a form of 3D porous material; these biomaterials consist of polymer chains with physical or chemical crosslinking [10]. Hydrogels can be of natural and/or synthetic origin [11]. Polyvinyl alcohol (PVA) hydrogels have been deeply explored due to their excellent biocompatibility properties and have been FDA (Food and Drug Administration) approved [12]. Hydrogels have received growing attention as drug delivery systems over the last decade due to their exclusive properties, such as high biocompatibility, tunable release rate, and versatility to be loaded with different molecules [13,14]. The most relevant characteristics of hydrogels are their porosity, pore size, and physicochemical environment of the matrix, which is tunable by modifying the crosslink density and/or varying the crosslinker type in their network. Therefore, the crosslinking agents such as aliphatic dicarboxylic acids (ADAs) could play a key role in the drug release profile from the hydrogel. In the hydrogel network, the pore size, swelling capacity, affinity with the drug, and subsequent sustained release have a very close relationship with the chemical structure and length of ADAs [14,15]. New antibacterial therapy strategies based on the sustained or prolonged release of linezolid could be an effective solution to fight pathogens. Thus, the purpose of this study was to develop hydrogels with ADAs of variable chain lengths as crosslinker agents featuring the tunable sustained release of linezolid.

## 2. Materials and Methods

### 2.1. Materials

Polyvinyl alcohol (PVA) 30–60 KDa, succinic acid (SA), glutaric acid (GA) adipic acid (AA), NaHCO_3_, acetonitrile (HPLC grade), and linezolid analytical standards were purchased from Sigma-Aldrich (St. Louis, MO, USA). HCl, methanol (HPLC grade), K_2_HPO_4_, and H_3_PO_4_ were purchased from Merck (Darmstadt, Germany). All solutions were prepared using MilliQ water. *Enterococcus faecium* ATCC^®^ 19434 bacterial strain, brain heart infusion (BHI) agar, Luria–Bertani (LB), and peptone water were purchased from Merck (Darmstadt, Germany). Distilled water was utilized for the preparation of all the solutions in the antibacterial study. The mouse fibroblast cell line L929 (ATCC^®^ CCL-1™) was purchased from ATCC (Manassas, VA, USA). The cells were cultured in Dulbecco’s modified Eagle’s medium (DMEM, Gibco^®^, Grand Island, NY, USA) containing 10% fetal bovine serum (FBS, Gibco^®^, Grand Island, NY, USA) and antibiotics (100 U penicillin and 100 U/mL streptomycin, Gibco^®^, Grand Island, NY, USA) under 5% CO_2_ at 37 °C. Cells were harvested after reaching confluence by using 0.05% trypsin–EDTA (Gibco^®^, Grand Island, NY, USA).

### 2.2. Synthesis of Hydrogels Based on PVA, ADAs, and Linezolid Loading

For this study, three hydrogels based on PVA and ADAs were synthetized. The methodology for preparing the hydrogels was performed through the esterification of PVA with ADAs according to the method from Rodríguez Nuñez et al. with minor modifications [16]. Briefly, the esterification reactions were performed by mixing an aqueous solution of PVA with an aqueous solution of a specific ADA (20 wt%) using HCl (1 × 10^−1^ mol·L^−1^) and temperature as catalysts of crosslinking. Then, each reaction was performed under reflux at ~ 90°C in a necked flask with magnetic agitation in the presence of air. After 3 h, each pre-hydrogel solution was poured into a new flask, and 8 mg of linezolid was added for its encapsulation, as depicted in Table 1. Then, each solution was vigorously stirred for 1 h and then sonicated for another hour until a homogenized solution was reached. After that, each pre-hydrogel-linezolid solution was put in an oven at 45 °C overnight until the crosslinking was complete. The hydrogel of PVA cross-linked with SA is named PSAH, the hydrogel of PVA cross-linked with GA named PGAH, and the hydrogel of PVA cross-linked with AA named PAAH. Afterward, the PSAH, PGAH, and PAAH hydrogels with the encapsulated linezolid were partially neutralized with NaHCO_3_ to remove the excess acid and increase water uptake [17]. Then, the linezolid-loaded hydrogels were lyophilized to obtain the xerogel. The linezolid-loaded PSAH, linezolid-loaded PGAH, and linezolid-loaded PAAH were termed PSAH-Li, PGAH-Li, PAAH-Li, respectively. At the same time, three hydrogels were prepared without linezolid to perform the following characterizations: ESR, FT-IR, TGA, mechanical analysis, and SEM.

### 2.3. Swelling Behavior

The swelling behavior was calculated by the equilibrium swelling ratio (% ESR) at desired time intervals. Each xerogel film was immersed in phosphate buffer saline (PBS) [18] and acetate buffer (pH 3.0) at 25 °C for 21 h until swelling equilibrium was attained. The weight of the wet sample (*W*_w_ (g)) was obtained after carefully eliminating moisture on the surface with an absorbent paper. The weight of the dried sample (*W*_d_ (g)) was acquired after freeze-drying the hydrogel sample. The ESR of the hydrogel samples was obtained as follows:(1)$$\mathrm{ESR}\;(\%)\;=\;\frac{W_{w}-W_{d}}{W_{d}}\times 100\%$$

### 2.4. FT-IR Analysis

The freeze-dried samples were ground into small fragments. After that, the PSAH, PGAH, and PAAH were analyzed in KBr (potassium bromide) disks by Fourier transform infrared spectroscopy (Nicolet Nexus 470 spectrometer, Thermo Scientific, Waltham, MA, USA). The wavenumber range scanned was 4000–500 cm^−1^; 32 scans of 2 cm^−1^ resolution were signal-averaged and stored. The films utilized in this analysis were sufficiently thin to obey the Beer–Lambert law.

### 2.5. Thermogravimetric Analysis

The thermal stability of PVA crosslinked films was evaluated using a thermobalance Cahn-2000 (Ventron Corp., CA, USA). Thermal analysis was carried out by heating samples (10 mg) from 25 to 600 °C at a heating rate of 10 °C/min under a nitrogen atmosphere (50 mL/min). The sample weight loss was recorded as a function of temperature.

### 2.6. Mechanical Properties

The tensile strength (TS), tensile modulus (E), and elongation at break (eB) of the hydrogels were measured according to American Society for Testing Materials (ASTM) D 882 test methods using an Autograph AGS-X Universal Tester (Shimadzu, Kyoto, Japan). The tensile samples were cut into rectangular shapes with dimensions of 100 mm in length and 10 mm in width. The gauge length was fixed at 50 mm, and the speed of the moving clamp was 5 mm·min^−1^. Three samples were tested, and the average values were taken as the reported results.

### 2.7. Scanning Electron Microscopy Analysis

Scanning Electron Microscopy (SEM) studies were carried out for all three formulations. The formulations morphology was evaluated using a scanning electron microscope (JEOL-JSM 6380, JEOL, Tokyo, Japan) operated at 15 kV. Surface views of cryogenically fracture films were examined. All samples were sputtered with a gold layer around 40 nm in thickness prior to the examination.

### 2.8. Release Kinetics Studies

The conformation of each proposed hydrogel is described in Table 1. Each linezolid-loaded hydrogel with a weight of 400 mg was placed into a 10 mL tube, and 5 mL of PBS [18] was poured over the formulation as a release medium. The tubes were transferred to an orbital shaking water bath (Faraz teb, Tehran, Iran) at 33.5 ± 0.1 °C [19] and shaken at 35 ± 2 rpm. At specific time intervals, the PBS was removed and replaced with an equal volume of PBS to maintain sink conditions throughout the study. For the quantification of linezolid, a stock solution (3 mg/mL) was prepared in methanol and stored at −18 °C. Standard solutions of the antibiotic were prepared with PBS (pH 7.4) in the range of 0.01–50 mg L^−1^. The chromatographic system consisted of a Perkin Elmer series 200 HPLC system (Norwalk, CT, USA) with a UV–vis detector and a C-18 Kromasil 100-5-C18 (250 mm × 4.6 mm i.d. × 5 μm) column. Fifty microliters of the sample were injected into the HPLC apparatus. Isocratic elution with methanol/water (50:50, *v/v*) at a constant flow rate of 1.0 mL·min^−1^ was utilized as the mobile phase. The analytical wavelength was 254 nm at room temperature The release rate of linezolid-loaded hydrogels was acquired by applying the concentration of released linezolid to the following correlation (Equation (2)):(2)$$\mathrm{Cumulative}\;\mathrm{Li}\;\mathrm{release}\;\left(\%\right)=\mathrm{Cumulative}\;\mathrm{amount}\;\mathrm{of}\;\mathrm{Li}\;\mathrm{released}\times \frac{100}{\mathrm{Inicial}\;\mathrm{amount}\;\mathrm{of}\;\mathrm{Li}}$$
Linezolid release kinetics were performed by employing different mathematical models of drug release equations, such as zero-order (Equation (3)), first-order (Equation (4)), Hixson–Crowell (Equation (5)), Higuchi (Equation (6)), Korsmeyer–Peppas (Equation (7)), and Peppas–Sahlin (Equation (8)) [20,21]:*Q_t_*/*Q*_0_ = *K*_0_*t*(3)
ln *Q_t_*/*Q*_0_ = *K*_1_*t*(4)
where *Q_t_* is the amount of linezolid released at time t, and *Q*_0_ is the original linezolid concentration in the formulation.
*C*_0_^1/3^ − *C_t_*^1/3^ = *Kt*(5)
where *C_t_* is the amount of drug released in time *t*, *C*_0_ is the initial amount of linezolid in the formulation, and *K* is the rate constant.
(6)$$Q=Kt^{1/2}$$
where *Q* is the cumulative linezolid release, *K* is the Higuchi release constant, and *t* is the time.
(7)$$\frac{M_{t}}{M}=Kt^{n}$$
where *M_t_/M* is the cumulative linezolid release, *K* is the release constant, *t* is the time, and n is the release exponent.
(8)$$\frac{M_{t}}{M\infty }=Kdt^{n}+Krt^{2n}$$
where *M_t_* and *M_∞_* are the absolute cumulative amounts of linezolid release at time *t* and at infinite time, respectively.

### 2.9. Antibacterial Activity

The studies were performed according to Oscar Forero-Doria et al. [8]. First, 50 mg of each linezolid-loaded hydrogel was placed into a tube with 5 mL of PBS [18] as “release medium”. Concurrently, a tube with 5 mL of PBS loaded with 1 mg of linezolid was prepared and utilized as a control. After that, the tubes were placed into an orbital shaking water bath (Farazteb, Iran) at 37 ± 0.1 °C. Depending on the assay, at certain time intervals (1, 3, 6, 24, and 48 h) 200 μL of release medium was taken and replaced with an equal volume of PBS to maintain sink conditions throughout the study. Lastly, the samples of each tube were evaluated by screening the antimicrobial activity and quantitatively testing the antibacterial activity utilizing the following protocols.

### 2.10. Assessment of Antimicrobial Activity of Proposed Hydrogels against E. faecium

To estimate the inhibitory activity against *E. faecium*, a qualitative test with a ring-diffusion method was implemented. With the aim of assessing the antibacterial activity of the prepared hydrogels, the Gram-positive strain *E. faecium* ATCC^®^ 19434 was used as a model pathogen. The bacteria were grown overnight in MRS (de Man Rogosa Sharpe) broth at 37 °C. The inoculum (100 μL) containing *E. faecium* (adjusted to ∼1.0 × 10^6^ CFU·mL^−1^) was seeded previously on the agar. Next, wells (8 mm in diameter) were made on an agar plate and filled with 100 μL of release medium for the specific interval times from Section 2.9. Additionally, two internal controls were treated with linezolid (10 and 15 μg·mL^−1^, respectively). The plates were incubated at 37 °C for 24 h, and the antibacterial activity was calculated by the formation of bacterial inhibition zones surrounding the film disks. All tests were performed in duplicate.

### 2.11. Quantitative Assay of the Antibacterial Activity of Proposed Hydrogels against E. faecium

In this analysis, *E. faecium* ATCC^®^ 19434 (concentration range of 1.0 × 10^6^ CFU·mL^−1^) was inoculated in 1 mL of LB broth at 37 °C until reaching turbidity equivalent to a 0.5 McFarland standard. Afterward, 150 μL of release medium from the samples and controls of Section 2.8 was added to 2 mL of the previous inoculation solution and then shaken at 200 rpm for 24 h at 37 °C. Afterward, each culture was tested; serial dilutions were made in 0.1% sterile peptone water. From each of these dilutions, 100 μL aliquots were collected, which were placed in plate count agar and incubated at 37 °C for 24 h. Then, viable cell counts were performed. All trials were performed in triplicate.

### 2.12. Cytotoxicity and Cell Viability

The cytotoxicity of the proposed hydrogels was evaluated on fibroblast cells. For this goal, the viability of the cells was assessed using the MTT technique according to the protocol of Mossman et al. [22]. Briefly, the cells were seeded in 24-well plates (5 μL, 1.6 × 10^4^ cells per well) and 150 μL of Dulbecco’s Modified Eagle Medium (DMEM)-High medium was added and incubated for 24 h at 37 °C in 5% CO_2_. Afterward, the medium was replaced by 100 μL of fresh DMEM-High per well, which containing three diverse concentrations of PSAH, PGAH, and PAAH (500 μg·mL^−1^, 1500 μg·mL^−1^, and 2500 μg·mL^−1^ per hydrogel). Fresh medium without a sample was utilized as a control. Cell viability was assessed after 24 h by the MTT technique. Briefly, 5 μL of MTT solution (3 mg·mL^−1^ in PBS) and 50 µL of fresh medium were added to the respective sample and incubated for 4 h in the dark at 37 °C; formazan crystals were then dissolved in 100 µL of DMSO and incubated for 18 h. Supernatant optical density (o.d.) was analyzed at 570 nm (Spectrophotometer, Packard Bell, Meriden, CT, USA). Unprocessed fibroblast cells were taken as control with 100% viability. The hydrogels cytotoxicity was depicted as the relative viability (%), which correlates with the number of viable cells compared with the negative cell control (100%).

### 2.13. Statistical Analysis

In this work, all experiments were performed in triplicate. The SPSS 9.0 statistical software (IBM, Chicago, IL, USA, 1999) was used to perform the ANOVA analysis and Tukey’s test (*p* < 0.05) to determine the statistical significance in some experiments such as the mechanical properties, ESR analysis, cumulative release test, quantitative test of antibacterial activity, and MTT assay. Graphs of the study results were designed by utilizing GraphPad Prism 6. Statistical significance was set at *p* < 0.05.

## 3. Results and Discussion

### 3.1. Synthesis and Load of Hydrogels

In Figure 1 the preparation of the hydrogels is depicted. Each hydrogel was prepared by esterification between PVA with a specific ADA (SA, GA, and AA). Once the pre-hydrogel was formed, the linezolid was added for its encapsulation (see Table 1). By this simple methodology, it is possible to achieve over 99% retention of the drug. Considering previous studies, a crosslinking degree of 10:2 of PVA:ADA was prepared, which was kept constant due to its good characteristics, such as porosity, mechanical properties, among others [14,16,23].

### 3.2. Swelling Behavior

This analysis allowed the confirmation of network formation in the three hydrogels. Since it is desirable to study the release of the antibiotic under physiological pH conditions, the ESR was evaluated at pH 7.4. Moreover, as a comparative analysis of swelling behavior, this analysis was also assessed at pH 4.0. Figure 2 shows the ESR for all formulations. An increase in the swelling index rate of the three hydrogels over time was observed at both pH values. In the beginning, the hydrogel swelling ratio increased fast and then slowed down to reach an equilibrium. This swelling behavior is characteristic of the hydrogel matrix obtaining the maximum swelling capacity. Specifically, PSAH, PGAH, and PAAH reached swelling equilibrium (zero-order) at approximately 3–4 h. On the other hand, a significant difference (*p* < 0.05) in all the cases was observed between the two pH models. For example, PSAH showed a better swelling degree at pH 7.4 with a value of approximately 600%, while at pH 4.0 the swelling degree was about 500%. For the other samples, PGAH and PAAH showed a swelling degree at pH 7.4 around 440% and 230%, and at pH 4.0 about 210% and 180%, respectively. The ESR difference at both pH values is due to the protonation degree of the free aliphatic carboxylic acids into the hydrogel network, which has different types of pKa [24]. Particularly, PGAH showed a higher difference in swelling behavior between pH values; such behavior can be attributed to the dissociation degree and ionization process of free COO- groups. These results are coherent, considering that the glutaric acid has the lowest pKa2 (5.22) compared to that of succinic and adipic acid (pKa2: 5.64 and 5.41), respectively. Therefore, in the network of PSAH more free carboxylic acid groups are susceptible to ionization by pH change.

When correlating the pH and Time variables for each formulation, it was found that for PSAH, the *p*-value < 0.05 (0.0140) in the ANOVA, therefore, there is a statistically significant relationship between the variables at the 95% confidence level. The highest *p*-value on the independent variables is 0.1175, belonging to pH. Thus, since the *p*-value> 0.05, pH is not statistically significant at the 95% or higher confidence level. In the case of PGAH, the *p*-value< 0.05 (0.0045) in the ANOVA, there is a statistically significant relationship between the variables at the 95% confidence level. On the other hand, the highest P-value on the independent variables is 0.0415 (*p*-value< 0.05), belonging to Time. In this context, Time is statistically significant at the 95% confidence level. Finally, in the case of PAAH, given that the *p*-value> 0.05 (0.1243) in the ANOVA, there is not a statistically significant relationship between the variables at the 95% or higher confidence level. The highest *p*-value on the independent variables is 0.9926 (*p*-value> 0.05), belonging to pH. Because of this, pH is not statistically significant at the 95% or higher confidence level. The correlation models are presented below:
PSAH correlation model:% ESR = 167.86 + 6.5042*t* (*R*^2^ = 35.85)(9)PGAH correlation model:% ESR = 44.5875 + 37.3718pH + 2.867*t* (*R*^2^ = 44.84)(10)PAAH correlation model:% ESR = 202.421 + 1.54335*t* (*R*^2^ = 14.17)(11)


### 3.3. FT-IR Analysis

The FT-IR analysis was conducted in the range from 4000 to 500 cm^−1^ to confirm the effectiveness of the crosslinking reaction between the PVA and different ADAs (SA, GA, and AA). Figure 3 shows the FT-IR spectra of hydrogel films (PSAH, PGAH, and PAAH). For PVA, all FT-IR spectra showed most of the characteristic infrared absorption bands (spectra not shown). The spectrum showed a broad band at around 3270 cm^−1^, which was attributed to inter- and intramolecular hydrogen bonds of -OH groups in PVA [25]. After crosslinking, this band showed a significant shift to 3400 cm^−1^ caused by the chemical reaction, demonstrating a polymer structural change. Other FT-IR bands appeared between 2840 and 3000 cm^−1^, around 1688 cm^−1^, and between 1150 and 1085 cm^−1^, corresponding to the vibrations of the -CH_2_, C=O, and C-O-C groups, respectively [26]. In addition, the evidence of ester formation between the PVA hydroxyl groups and diacids carboxylic groups is the shifting of the ADA’s -C=O absorption peak from 1691 to 1704 cm^−1^. This result was in agreement with that previously reported in [27]. It the intensity difference for the -CH stretching vibration (between 2840 and 3000 cm^−1^) between crosslinked films could also be noticed. The band intensity increased along with the crosslinker agent carbon numbers, as shown in Figure 1. It is important to note that other significant absorption bands recorded for PSAH, PGAH, and PAAH samples demonstrated the acid compounds in the hydrogel structure [14]. Table 2 summarizes the leading characteristics bands and their assignment for neat PVA and hydrogel samples. Finally, the spectral changes obtained in FT-IR analysis demonstrated the success of the crosslinking reaction between the hydroxyl group of PVA and the carboxylic groups from ADAs.

### 3.4. Thermogravimetric Analysis

Thermogravimetric (TG) analysis has become a frequently used technique for studying the thermal stability of complex materials. In this work, the thermal properties of the prepared hydrogels based on PVA crosslinked with different ADAs are investigated using TG. The TG and Derivative thermogravimetry (DTG) curves of hydrogel samples are presented in Figure 4, and the analysis results are summarized in Table 3.

It is known the degradation of neat PVA is observed over three temperature regions, which are 80–250 °C, 275–450 °C, and 475–525 °C peaking at 142, 287, and 440 °C, respectively [27]. For all formulations, a first thermal effect with a maximum decomposition rate from 137 to 142 °C and an associated mass loss from 8.8% to 12.7% were observed. This effect is related to the evaporation of the bound and unbound water of the films [25]. The thermal degradation of crosslinked films occurred in three temperature regions for PGAH and PAAH, but in four regions for PSAH. The second degradation step showed two stages for PSAH; the first T_Peak_ appeared at 224 °C, and the second at 297 °C. The first process could be due to the thermal degradation of the free SA remaining in the hydrogel and showed a weight loss of 8.8%, whereas the second loss was 17.8%. In the case of PGAH and PAAH, the second effect was peaking at 245 and 273 °C with associated weight losses of 12.5% and 27.9%, respectively. All those peaks could correspond to a shift in PVA decomposition temperature (287 °C) (data not shown). In this thermal effect, the scission of partially esterified but still uncrosslinked PVA chains were co-occurring with polymer cyclization [27]. The third decomposition stage belonged to crosslinked PVA chain decomposition. The maximum decomposition rate appeared from 345 to 369 °C, demonstrating an increase in the thermal stability of formed hydrogels. Due to the formation of multiple inter- and intrachain ester bonds, an interpolymeric network that modifies the PVA structure was created. After crosslinking, an increase in the number of covalent bonds and hydrogen bonding between the polymer chains occurred, making them more thermally stable. This finding agrees with results from other authors that crosslinked PVA with AA and GA [28]. This thermal effect was more relevant for the PGAH sample, showing 39.7% of weight loss. The last thermal effect showed a maximum decomposition rate in the temperature interval of 435–449 °C and mass loss associated from 27.8% to 34.8%. Several authors reported that in this process the complete degradation of the PVA backbone occurs and cyclized chains turn them into charring residue [25,27].

### 3.5. Mechanical Properties

The influence of the ADA’s chain length on the mechanical properties of crosslinked PVA hydrogel films is summarized in Table 4. Tensile strength (TS), tensile modulus (E), and elongation at break (eB) were included in the evaluated tensile properties.

The results showed interesting mechanical properties of the hydrogel films prepared with ADAs. We hypothesized that the formation of the ester bond between the ADAs and the PVA matrix would play an important role in the mechanical properties, dependent on the chain length of the ADAs used. Thus, the PGAH hydrogel exhibits better mechanical properties (TS: 25 MPa; eB: 126%) than the PSAH and PAAH hydrogels. It was significantly different in TS values with the PSAH hydrogel and for eB values with the PSAH and PAAH samples, respectively. On the other hand, the PAAH hydrogel showed a significant difference in E values compared to the other ones, which means it is the stiffer sample. This result could be due to the degree of crosslinking obtained between the PVA matrix and glutaric acid chains during the formation of the hydrogel network. This fact agrees with the SEM analysis (Figure 5) of this sample, which showed a rough and homogeneous surface that improves mechanical strength. The mechanical properties of the materials are associated with their chemical nature and the interactions among the forming components [29]. The results reveal that crosslinking agents with intermediate chain lengths (GA) favor a higher chemical interaction with the PVA matrix.

Moreover, thermal analysis results confirm this finding because a higher crosslinking degree was found in this sample. Furthermore, previous hydrogels based on PVA found a linear relationship between mechanical strength and the crosslinking degree [30]. These results are in agreement with the FT-IR analysis regarding the ester band appearance around 1704 cm^−1^. On the other hand, the PAAH hydrogel showed mechanical properties similar to the stiff and brittle materials with high tensile modulus and low elongation at break. This behavior could be associated with the porosity and the highly compact structure found in the fractured surface of hydrogel films observed by the SEM (Figure 5). The lower mechanical behavior of the hydrogel films prepared with PAAH was associated with the crosslinker characteristics, consisting of a larger molecule that restricts the formation of crosslinking density in the hydrogel [31]. The tensile strength and tensile modulus indicate the toughness, and the elongation at break indicates the elasticity of the materials, suggesting their possible applications. The hydrogels of this study can be used for drug delivery applications such as antibiotic release because they have adequate (strong and flexible) mechanical properties [32,33]. Thus, several authors [34,35] reported the following values for the mechanical properties of wound dressing hydrogel films (TS = 18 MPa; E = 98; eB = 200%) and drug delivery films (TS = 13–35; eB = 44–112%), respectively.

### 3.6. SEM Analysis

The hydrogel formulations with different crosslinkers were observed using Scanning Electron Microscopy (SEM) to understand surface properties. As depicted in Figure 5, for all samples, a rough surface is observed. In the case of PGAH (Figure 5B), a more uniform structure than those in the PSAH sample could be observed (Figure 5B). Finally, the PAAH micrograph (Figure 5C) revealed a different and more compact morphology with hollows fractures on the surface. It seems that in hydrogel formulations, the surface morphology was highly influenced by the crosslinker chemical structure, being stiff and dense when the ADA’s chain length is longer. Then the hydrogel supramolecular structure could affect the drug release behavior.

### 3.7. Release Kinetics Studies

The linezolid release profile analysis was carried out by HPLC with a 400 mg hydrogel charged with linezolid. In vitro release kinetics of linezolid from each hydrogel were obtained under physiological conditions (37 °C, PBS at pH 7.4). The cumulative percent released of the antibiotic was monitored over time and results are shown in Figure 6.

For all loaded hydrogels (PSAH-Li, PGAH-Li, and PAAH-Li) a rapid antibiotic release into the medium was observed. The cumulative release of PSAH was significantly higher than the other samples from 2 h. From 6 h onward, all hydrogels showed a significantly different drug release profile. At this time, 51%, 40%, and 29% of the linezolid had been released from PSAH-Li, PGAH-Li, and PAAH-Li, respectively. The PAAH-Li revealed a slower release profile than the other two formulations, and PSAH-Li a higher one. After 6 h, the formulations exhibited a significantly lower and continuous antibiotic release into the medium. For PSAH-Li, PGAH-Li and PAAH-Li, the average rapid-release phase was 0.68, 0.53, and 0.39 mg/h of linezolid, respectively. This rate changed after 6 h for all cases, and the average of the slow-release phase was 0.06, 0.05, 0.04 mg/h of linezolid, respectively. In Table 5, all the average release values of the antibiotic are shown. According to these results and the graph depicted in Figure 6, the linezolid release profile follows the next order: PSAH-Li > PGAH-Li > PAAH-Li.

The release patterns of each formulation depend on the structure of the crosslinker agent, the intermolecular interactions between the linezolid drug and hydrogel network [12], and mostly on the swelling degree. Therefore, in light of the obtained outcomes, it could be deduced that the release rate of PAAH-Li was slower because the expansion of the compact network was minimal, as revealed by the water uptake process (% ESR). One of the reasons for this result is that the size of AA aliphatic chain is larger than SA and GA, respectively, which could contribute to a more apolar environment. Moreover, in concordance with the mechanical studies, PAAH showed a stiffer structure with high tensile modulus and low elongation at break. This performance has a direct relation with its highly compact morphology observed by SEM. The lower mechanical behavior of PAAH could be associated with the larger crosslinker agent that limits the formation of crosslinking density in the hydrogel. With a lesser crosslinking degree, there are less carboxylic groups potentially ionizable. Therefore, there are fewer charges of the electrostatic repulsion between chains from networks and, consequently, less capacity to generate an uptake of solvent into the matrix. This result is consistent with swelling behavior. On the contrary, for the case of PSAH and PGAH with a higher crosslinking degree, the increased swelling ability of the hydrogels contributes to the destruction of hydrogen bonding between the polymer molecules, resulting in an increase in chain mobility and network expansion [21]. The above mentioned results can explain the faster release of linezolid from PSAH-Li. In contrast, in a lesser expanded, more compacted, and stiff hydrogel (PAAH-Li), the encapsulated drug is released slower.

The average release profiles of samples were fitted through several mathematical models to elucidate the mechanism of linezolid release. The coefficients of correlation (*R*) and release exponents (*n*) are shown in Table 6. According to *R^2^* value obtained, among all the studied models, the Korsmeyer–Peppas model was the best fit for PSAH-Li and PGAH-Li with *R^2^* values of 0.9967 and 0.9845, respectively. In contrast, an with *R^2^* value of 0.9013, the Higushi model is the best fitted to PAAH-Li. On the other hand, for the case of PSAH-Li and PGAH-Li the release mechanism for linezolid was Fickian diffusion. For the case of PAAH-Li the mechanism for linezolid release was pseudo-Fickian.

### 3.8. Antibacterial Studies

Some studies indicate that the effectivity of antibacterial agents is better when they are applied through sustained release. Furthermore, this approach could lower bacterial resistance incidence [7,8,36]. Regarding the limitations presented by PSAH (faster release) and PAAH (reduced mechanical properties), PGAH was selected to carry out the antibacterial analyses. The results obtained here are consistent with the acquired data by HPLC. For instance, linezolid antibacterial activity was significantly higher over time compared to the control, as shown in Figure 7. In this context, it is concluded that a sustained release at a relatively constant dose from PGAH-Li maintains antibiotic integrity, comprising better activity over time, as shown in Figure 7B,C. On the contrary, in the control sample, linezolid displayed higher activity in the first hour; however, it loses effectiveness over time, as depicted in Figure 7A,C. The control inhibition zone in the first hour was close to ~30 mm, but over time was decreasing, reaching ~8 mm at 48 h. In contrast, PGAH-Li started with an inhibition zone of ~11 mm and progressively rose to complete an inhibition zone of 23 mm at 48 h. These results are in concordance with the quantitative analysis of antibacterial activity against the *E. faecium* that is exhibited in Figure 7D. The data revealed that, in the control, the bacterial colony forming unit (CFU) increases over time, demonstrating that the antibiotic loses its activity until 72 h. Conversely, the assay with the PGAH-Li release medium significantly inhibits bacterial proliferation, suggesting that the linezolid acted as a bacteriostatic agent against *Enterococcus faecium* [37]. An additional experiment with PGAH without linezolid was performed. However, the antibacterial activity was not observed (data not shown). These data suggest that the hydrogel could improve the bioavailability of linezolid.

### 3.9. Cytotoxicity Studies

As potential biomaterials, it is pivotal that the designed formulations be innocuous. Therefore, the cytotoxicity of each hydrogel was evaluated on fibroblast cells. The biocompatibility of the sterilized PSAH, PGAH, and PAAH was investigated by a cell viability assay using L929 fibroblast cells after 24 h. Figure 8 displayed cell viability after exposure to three different concentrations of the respective formulation (a concentration range of 500–2500 μg·mL^−1^). As specified in Figure 8, at 500 μg·mL^−1,^ the fibroblast cell viability is nearly 100% for three hydrogels. When drastically increasing the hydrogel concentration from three to five-fold, the fibroblast cell viability only declines vaguely. That is to say, for all cases, the cell viability was not less than 87%. The results revealed that the prepared hydrogels have minimum toxicity to the fibroblast cell. These data could guarantee that these hydrogels can be potential candidates for medical applications.

## 4. Conclusions

A series of hydrogels based on PVA and ADAs of variable chain lengths with sustained release of linezolid properties were successfully synthesized. The hydrogels were prepared by crosslinking of PVA and different ADAs of varying chain lengths, such as SA, GA, and AA, respectively. The swelling response, FT-IR, TGA, mechanical properties, and SEM analysis validate the formation of the three hydrogels. The swelling index data evidenced that all the proposed hydrogels are responsive to pH. Moreover, the swelling index depends on the type of ADA and crosslinking degree. The series of hydrogels showed a sustained release rate of linezolid according to the results shown in the chromatographic analysis. The three hydrogels displayed significant differences regarding the release rate of linezolid. This difference seems to be ruled by the intermolecular interactions between linezolid and hydrogel matrix morphology, crosslinking degree, and mechanical properties. These mentioned features have a direct relation with ADA type used as crosslinker. ADAs can confer unique physicochemical and mechanical properties based on their specific structure. Therefore, the ADAs could play a key role in the release profile of the drug. The linezolid release kinetic of PSAH-Li and PGAH-Li were found to follow the Korsmeyer–Peppas release model, and the release mechanism in both cases was Fickian diffusion. On the contrary, the Higushi model was the best fit for PAAH-Li, and their mechanism for linezolid release was pseudo-Fickian. The antibacterial assays confirmed that the sustained release of linezolid from PGAH-Li has a better antibacterial activity compared with the conventional release. This suggests that the hydrogel has the capability to improve the bioavailability of linezolid. The set of proposed hydrogels showed good biocompatibility with L929 mouse connective tissue fibroblasts. The results exhibited viability over 87%. In conclusion, drug delivery platforms based on hydrogels of PVA and specific crosslinker agents such as ADAs could be potentially utilized as an antibiotic delivery system in potential infectious processes. Furthermore, this approach could become a strategy to help stop bacterial resistance.

## Figures and Tables

**Figure 1 pharmaceutics-12-00982-f001:**
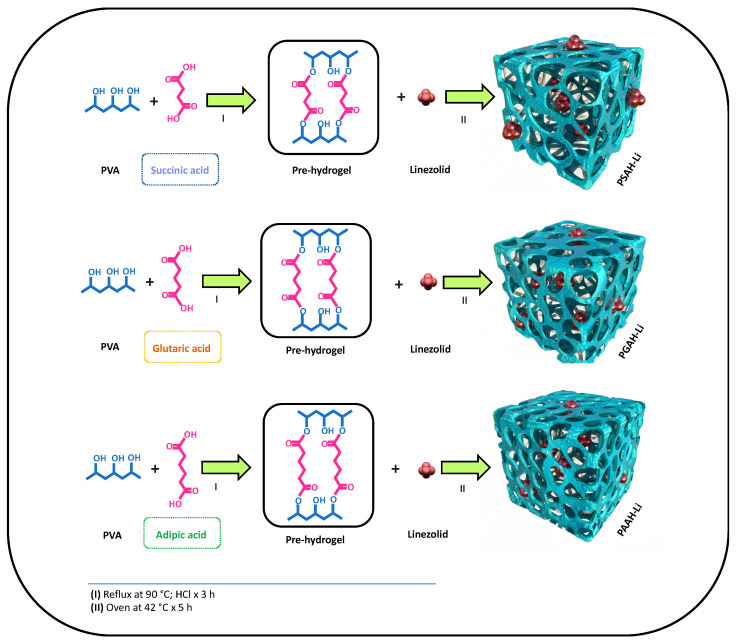
Synthesis and proposed structures of the linezolid-loaded hydrogels.

**Figure 2 pharmaceutics-12-00982-f002:**
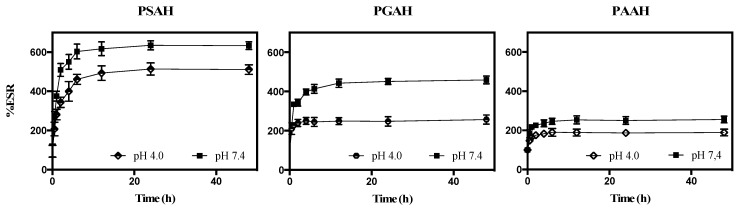
The swelling ratio of the hydrogels at 24–25 °C as a function of time, pH, and crosslinker nature. Data are shown as mean ± SD (*n* = 3).

**Figure 3 pharmaceutics-12-00982-f003:**
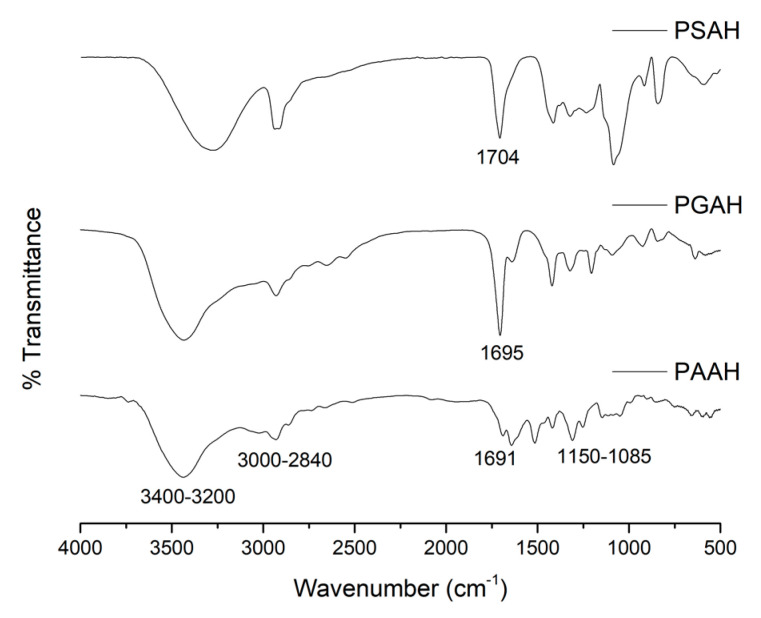
FT-IR spectra of PSAH, PGAH, and PAAH.

**Figure 4 pharmaceutics-12-00982-f004:**
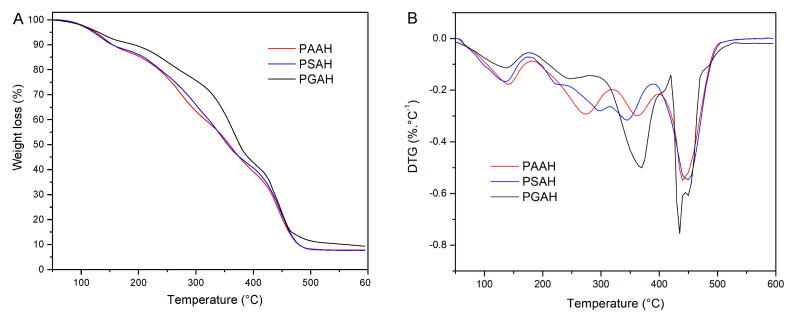
TG (**A**) and DTG (**B**) curves of the PSAH, PGAH, and PAAH recorded at 10 °C/min in the N_2_ atmosphere.

**Figure 5 pharmaceutics-12-00982-f005:**
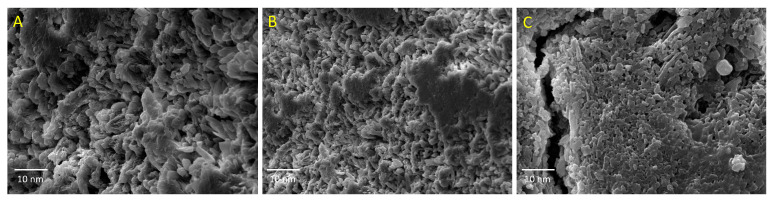
SEM micrographics of PSAH (**A**), PGAH (**B**), and PAAH (**C**).

**Figure 6 pharmaceutics-12-00982-f006:**
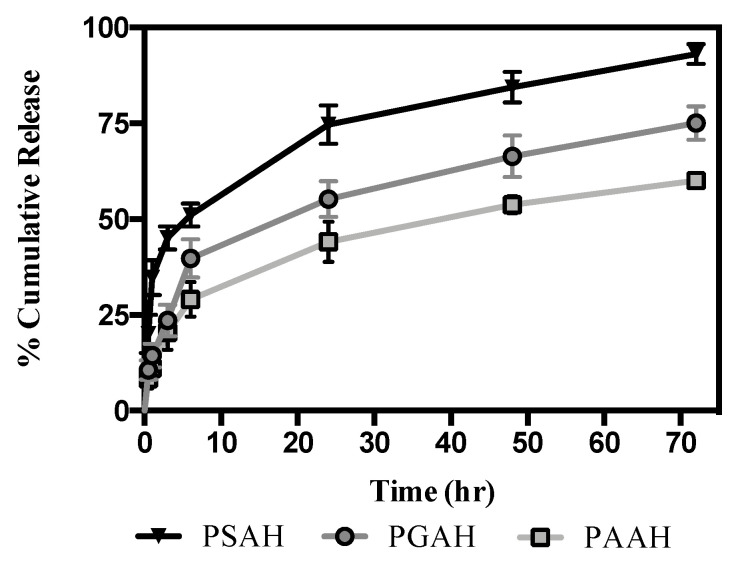
Release profile of linezolid from prepared hydrogels in phosphate buffer saline (PBS) at 33.4 °C; mean Scanning Electron Microscopy (SEM) (*n* = 3). Different letters next to the standard deviation on each point indicate statistically significant differences using Tukey HSD, at 95% confidence.

**Figure 7 pharmaceutics-12-00982-f007:**
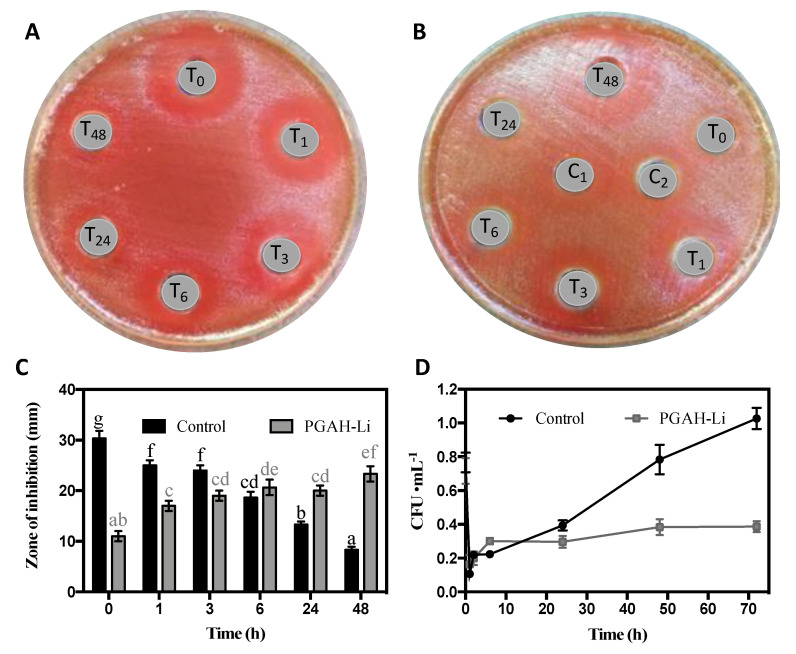
Screening of the antibacterial effect of PGAH-Li. Control (**A**); PGAH-Li (**B**); T_0_: 0 h, T_1_: 1 h, T_3_: 3 h, T_4_: 6 h, T_24_: 24 h, T_48_: 48 h; the antibacterial effect was expressed as the inhibition area against *E. faecium* (**C**); quantitative test of antibacterial activity against *E. faecium* (**D**). (Equal letters above the bars indicate that there are no statistically significant differences using Tukey’s HSD procedure, at 95% confidence level). C_1_ and C_2_ in Figure 7B are positive controls of 15 and 10 μg·mL^−1^ linezolid, respectively.

**Figure 8 pharmaceutics-12-00982-f008:**
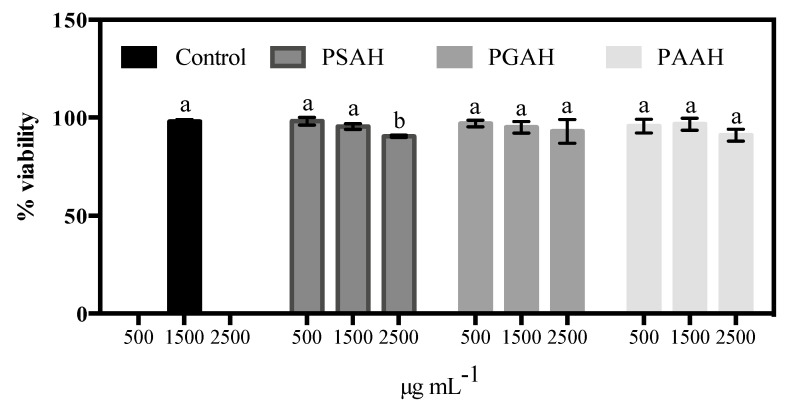
Percentage of cell viability obtained from the MTT assay of the L929 fibroblast cells compared to that of a negative control (without hydrogel). Each bar indicates mean ± relative standard deviations (RSD) of three replications. Bars not labeled by the same letter represent statistically significant differences with the negative control at *p* ≤ 0.05 using ANOVA followed by Tukey’s HSD test.

**Table 1 pharmaceutics-12-00982-t001:** Specifications of prepared hydrogels and the amount of linezolid loading.

Hydrogel	Crosslinker	Crosslinker Ratio (%) *	Linezolid (%) *
PVA-Succinic Acid (PSAH)	SA	20	2
PVA-Glutaric Acid (PGAH)	GA	20	2
PVA-Adipic Acid (PAAH)	AA	20	2

* % *w/w* respect to hydrogel; succinic acid (SA); glutaric acid (GA), and adipic acid (AA).

**Table 2 pharmaceutics-12-00982-t002:** Vibration modes and band frequencies in polyvinyl alcohol (PVA) and PVA crosslinked with SA, GA, and AA.

Sample	Chemical Group	Wave Numbers (cm^−1^)
PVA PSAH PGAH PAAH	O-H from the intermolecular and intramolecular hydrogen bonds	ν ~3400
PVA PSAH PGAH PAAH	C-H from alkyl groups	ν 2840–3000
PSAH PGAH PAAH	C=O	ν 1704 ν 1695 ν 1691
PVA	-C=C	ν 1640
PVA	CO (crystallinity)	ν 1100
PVA PSAH PGAH PAAH	C-O-C	ν 1150–1085
PVA PSAH PGAH PAAH	CH_2_	δ 1461–1417

**Table 3 pharmaceutics-12-00982-t003:** Thermogravimetric analysis results from hydrogels.

Sample	Temperature (°C)			Weight Loss (%)	Char (%)
	Onset	Peak	End		
**PSAH**	36.5	137.6	175.1	11.8	7.4
	177.5	224.2	240.9	8.5	
	241.0	297.2	316.9	17.8	
	317.0	345.8	390.6	19.5	
	391.0	449.9	528.9	34.8	
**PGAH**	36.1	138.7	174.8	8.8	9.3
	175.0	245.4	281.9	12.5	
	289.5	369.4	420.3	39.7	
	420.7	435.4	528.1	27.8	
**PAAH**	35.4	142.8	181.6	12.7	7.6
	182.4	273.6	319.1	27.9	
	319.2	363.5	399.7	20.1	
	400.0	440.9	530.9	31.3	

**Table 4 pharmaceutics-12-00982-t004:** Mechanical properties of PVA crosslinked films *.

Formulation	Tensile Modulus (E) (MPa)	Tensile Strength (TS) (MPa)	Elongation at Break (eB) (%)
PSAH	80 ± 10 a	14 ± 1 a	88 ± 9 a
PGAH	70 ± 7 a	25 ± 5 b	126 ± 26 b
PAAH	104 ± 4 b	17 ± 4 b	66 ± 13 a

* Different letters next to the standard deviation, in each column, indicate statistically significant differences using Tukey HSD (honestly significant difference), at 95% confidence.

**Table 5 pharmaceutics-12-00982-t005:** Release profile of linezolid-loaded hydrogels.

Formulation			PSAH-Li	PGAH-Li	PAAH-Li
Release phase	Rapid *	% Released	51 ± 3 c	40 ± 5 b	29 ± 4.6 a
		Release rate (mg/h)	0.68 ± 0.04 b	0.53 ± 0.07 a	0.39 ± 0.06 a
	Slow **	% Released	42 ± 3.6 b	35.3 ± 3.5 ab	30.3 ± 3.5 a
		Release rate (mg/h)	0.06 ± 0.005 b	0.05 ± 0.005 ab	0.04 ± 0.005 a

* The rapid phase occurred over 6 h. ** The release rate was calculated in a specific time frame because, until 72 h, the formulation still released antibiotic. Different letters next to the standard deviation, in each row, indicate statistically significant differences using Tukey HSD, at 95% confidence.

**Table 6 pharmaceutics-12-00982-t006:** Linezolid release kinetics and correlation coefficient values from Fick, Hixon–Crowell, Higushi and Korsmeyer–Peppas models.

Hydrogel	Mathematical Model										
	Zero Order		First Order		Hixon-Crowell		Higushi		Korsmeyer-Peppas		
	*R* ^2^	*K*	*R* ^2^	*K*	*R* ^2^	*K*	*R* ^2^	*K*	*R* ^2^	*K*	*n*
PSAH-Li	0.8382	1.0054	0.6736	0.0227	0.4724	−0.0324	0.9632	3.4568	0.9967	5.7570	0.5112
PGAH-Li	0.8326	1.2624	0.6678	0.0225	0.4671	−0.0348	0.9529	4.3222	0.9845	7.3148	0.5167
PAAH-Li	0.7585	1.6056	0.6558	0.0159	0.3590	−0.0324	0.9013	4.8923	0.8938	14.6218	0.3539

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
