# Peer review of "Sustained Release of Linezolid from Prepared Hydrogels with Polyvinyl Alcohol and Aliphatic Dicarboxylic Acids of Variable Chain Lengths"

_pharmaceutics, 2020, doi:10.3390/pharmaceutics12100982_

Round 1

Reviewer 1 Report

Authors are describing a hydrogel platform made of cross-linked PVA potentially useful for drug delivery. The manuscript is overall well written and clear. Nevertheless there are some minor comments that I believe will improve the manuscript. 

1. There are not statistics in the whole paper, without them is really hard to make any conclusion with these results.

2. Could authors run also PVA in the experiments ( FITR, DSC and TGA). It could be helpful. Also, adding arrows in the images referring to the peaks they are talking about will help a lot to the reader.

3. Images don’t have a scale. I cannot see that many differences in porosity with these images.

Author Response

Response to the reviewer 1

Authors are describing a hydrogel platform made of cross-linked PVA potentially useful for drug delivery. The manuscript is overall well written and clear. Nevertheless there are some minor comments that I believe will improve the manuscript.

Response: We thank the reviewer for their appreciation and comments. The changes associated with the comments and suggestions of the reviewer were marked in yellow in the manuscript

  1. There are not statistics in the whole paper, without them is really hard to make any conclusion with these results.

Response: Regarding the opinion of the reviewer on the lack of statistics in this work, some clarifications are attached:

- In Table 5, according to the reviewer's recommendations, the analysis of statistically significant differences was performed for the release profile, comparing the different prepared hydrogels.

- Figure 2 includes measurements in triplicate, so the standard deviation associated with each point of the graphs comparing the swelling ratio of hydrogels as a function of time, pH, and crosslinker type are included. Since the standard deviations associated with each measurement do not overlap with the results of measurements at different pHs for each hydrogel separately, it is not necessary to do tests of statistical significance, such as LSD, Tukey HSD, among others. The conclusions from observing the graphs are quite clear and simple for the objective reader.

- Figure 8 includes a Tukey HSD test to determine the statistically significant differences in cell viability with respect to hydrogels.

  1. Could authors run also PVA in the experiments ( FITR, DSC, and TGA). It could be helpful. Also, adding arrows in the images referring to the peaks they are talking about will help a lot to the reader.

Response: We thank the referees for the suggestions made to improve the discussion of the FITR, DSC, and TGA curves for the obtained hydrogels. However, in this research, PVA is one of the reactants used in the formation of hydrogels. As can be seen, all the data discussion referred to the characterization of the synthesized hydrogels. Moreover, taking into account that PVA is widely referenced in the literature and the high volume of data shown in this research, we decided not to provide the data referring to the characterization of PVA. Despite this, in the new version of the manuscript, the main values of the signals were added in Figure 3 according to the referee’s recommendation.

  1. Images don’t have a scale. I cannot see that many differences in porosity with these images.

Response: The scale was added to the images in Figure 5. On the other hand, the images were processed and repeat several times looking for better differences in porosity, however, it wasn’t possible.

Reviewer 2 Report

Dear authors,

you performed an interesting research. Congrats.

I have some suggestions for Manuscript improvement.

All my comments are listed below with an appropriate Line number(s) from text in order to facilitate tracking.

General comments:

  1. There is an error regarding to citing of 2 or more references in text. According to MDPI rule there is no need for "space" between numbers of cited references. For instance, it should be (1,2) instead of (1, 2). Please correct this issue through a whole Manuscript i.e. in Lines 49, 61, 67,73, 418.
  2. When you are citing some reference in form "author et al." there is no need for year and comma after that. In that way please correct the following references in text as follow: Line 95 ("Rodriguez Nunez et al. (16)"), Line 179 ("Forero-Doria et al. (8)"), Line 232 ("Avila Salas et al. (14)"), Line 353 ("Kord et al. (33) and Singh et al. (34)").

Specific comments:

Line 36: I think it should be "were investigated" not "was investigated" since you are talking about "release kinetics studies".

Line 42: Suggest to authors to include in Abstract some numerical value(s) for this "constant doses".

Line 44: "polyvinyl alcohol" instead of "polyvinylalcohol". The same error in Lines 80.

Lines 94-102: Suggest to authors to give more details about polymer synthesis i.e. did you use some specific initiatior for polimerization, did you perfrom reaction with or withouth presence of air/oxygen? In addition, I think that part of text (Lines 231-238) from the begining of Results section is much more appropriate for Material and methods section. So, I suggest to authors to move it there.

Line 103: Did you perform the complete neutralisation with NaHCO3 or partial? Sometimes hydrogels express better swelling properties when neutralisation is 60% instead of 100%. Precise here.

Line 125: Split numerical value (10) from unit (mg).

Line 150: Please separate number "50" from "-".

Lines 257-259: I could not agree in total here with you. If you have higher crosslinking ration than you can not have higher swelling process since they are oposite to each other. I think that here is more important disociation degree and ionisation process of free COO- ions. And since glutaric acid has the lowest pKa2 value (5.22= compared to the other ones (5.64 for succinic and 5.41 for adipic acid) it is logical to me that it can abosrb more water at pH 7.4 compared to pH= 4.0. Please check rephrase this part of text.

Line 328: Please correct technical errors in Table 4. Decimal numbers should be given as in English alphabet i.e. for instance 79.5 instead of 79,5.

Line 359: Reorder as follow: "... relatively uniform pores".

Line 360: "is smaller" not "are smaller"?

Line 381: Split words "on the".

Lines 380-384: If you have slower release than you have smaller pores in hydrogels? Your SEM analysis that PAAH had small pores. So, your conclusion in Lines 382-384 is not in Line with your experimental data. Please check and correct adequately.

Line 433: I think that "Enterococcus" should be with capital letter E and in Italic?

Line 435: Delete "to" after "could". It is surplus here.

Author Response

Response to the reviewer 2

Response: We thank the reviewer for appreciation and comments. The changes associated with the comments and suggestions of the reviewer were marked in yellow in the manuscript

General comments:

  1. There is an error regarding to citing of 2 or more references in text. According to MDPI rule there is no need for "space" between numbers of cited references. For instance, it should be (1,2) instead of (1, 2). Please correct this issue through a whole Manuscript i.e. in Lines 49, 61, 67,73, 418.

Response: According to the suggestion of the reviewer the references were fixed.

  1. When you are citing some reference in form "author et al." there is no need for year and comma after that. In that way please correct the following references in text as follow: Line 95 ("Rodriguez Nunez et al. (16)"), Line 179 ("Forero-Doria et al. (8)"), Line 232 ("Avila Salas et al. (14)"), Line 353 ("Kord et al. (33) and Singh et al. (34)").

Specific comments:

Response: According to the suggestion of the reviewer the references were fixed

  1. Line 36: I think it should be "were investigated" not "was investigated" since you are talking about "release kinetics studies".

Response: Line 36, the sentence was corrected

  1. Line 42: Suggest to authors to include in Abstract some numerical value(s) for this "constant doses".

Response: Line 40, according to the suggestion of the reviewer the sentence was modified

  1. Line 44: "polyvinyl alcohol" instead of "polyvinylalcohol". The same error in Lines 80.

Response: According to the suggestion of the reviewer, this error was corrected across the text.

  1. Lines 94-102: Suggest to authors to give more details about polymer synthesis i.e. did you use some specific initiatior for polimerization, did you perfrom reaction with or withouth presence of air/oxygen? In addition, I think that part of text (Lines 231-238) from the begining of Results section is much more appropriate for Material and methods section. So, I suggest to authors to move it there.

Response: Line 101-105,  according to the suggestion of the reviewer, the modifications were performed

  1. Line 103: Did you perform the complete neutralisation with NaHCO3 or partial? Sometimes hydrogels express better swelling properties when neutralisation is 60% instead of 100%. Precise here.

Response: Line 108-110, the methodology of neutralization performed was clarified. 

  1. Line 125: Split numerical value (10) from unit (mg).

Response: Line 137, the numerical valued was corrected. 

  1. Line 150: Please separate number "50" from "-".

Response: Line 162, the error was corrected.

  1. Lines 257-259: I could not agree in total here with you. If you have higher crosslinking ration than you can not have higher swelling process since they are oposite to each other. I think that here is more important disociation degree and ionisation process of free COO- ions. And since glutaric acid has the lowest pKa2 value (5.22= compared to the other ones (5.64 for succinic and 5.41 for adipic acid) it is logical to me that it can abosrb more water at pH 7.4 compared to pH= 4.0. Please check rephrase this part of text.

Response: Line 280-285. According to the suggestion of the reviewer, the sentence was rephrased. Thank you very much for this observation.

  1. Line 328: Please correct technical errors in Table 4. Decimal numbers should be given as in English alphabet i.e. for instance 79.5 instead of 79,5.

Response: According to the suggestion of the reviewer, the Table 4 was corrected. 

  1. Line 359: Reorder as follow: "... relatively uniform pores".

Response: Line 406, the sentence was rewritten  

  1. Line 360: "is smaller" not "are smaller"?

Response: Line 407, the word was changed

  1. Line 381: Split words "on the".

Response: Line 431, the error was corrected

  1. Lines 380-384: If you have slower release than you have smaller pores in hydrogels? Your SEM analysis that PAAH had small pores. So, your conclusion in Lines 382-384 is not in Line with your experimental data. Please check and correct adequately.

Response: Line 431-435. According to the suggestion of the reviewer, the sentence was rewritten. Thank you very much for this observation.

  1. Line 433: I think that "Enterococcus" should be with capital letter E and in Italic?

Response: Line 490, the error was corrected.

  1. Line 435: Delete "to" after "could". It is surplus here.

Response: Line 492, the error was corrected.

Round 2

Reviewer 1 Report

  1. There are not statistics in the whole paper, without them is really hard to make any conclusion with these results. Still missing some statistics: in figure 2 and 7 table 3 and 4. And although I agree that they are significative different, but still need to be tested, especially when affirming that they are differences between samples.

  1. If authors didn’t find any SEM image showing the differences in porosities, they cannot state that they are differences.

"In the case of PGAH (Fig. 5B), a rough and porous surface is observed which is smaller and more uniform than those in the PSAH sample. Finally, the PAAH (Fig. 5C) revealed a stiff and more compact structure with pores smaller than the other two hydrogel samples. It seems that in the formulations the surface morphology was highly influenced by the crosslinker chemical structure"

Author Response

Response to the reviewer 1- Round 2

Response: We thank the reviewer for appreciation and comments. The changes associated with the comments and suggestions of reviewer were marked in green in the manuscript

Comments and Suggestions for Authors

  1. There are not statistics in the whole paper, without them is really hard to make any conclusion with these results. Still missing some statistics: in figure 2 and 7 table 3 and 4. And although I agree that they are significative different, but still need to be tested, especially when affirming that they are differences between samples.

Response: Regarding the statistical analysis in Table 3, it does not make sense because the TGA data always comes from single sample analysis. On the other hand, the data in Table 4 were analyzed according to the Tukey test, and the results were placed in the corresponding table. Besides, the following sentences were included in the text from lines 341 to 343 (Highlighted in green) “It was significantly different in TS values with PSAH hydrogel and for eB values with PSAH and PAAH samples, respectively. On the other side, the PAAH hydrogel showed a significant difference in E values regarding the other ones, which means it is the stiffer sample”.

On the other hand, in Figures 2 and 7 the statistics analysis was performed. In Figure 2, the test ANOVA was applied. In Figure 7, the tukey's HSD test was applied.

  1. If authors didn’t find any SEM image showing the differences in porosities, they cannot state that they are differences.

"In the case of PGAH (Fig. 5B), a rough and porous surface is observed which is smaller and more uniform than those in the PSAH sample. Finally, the PAAH (Fig. 5C) revealed a stiff and more compact structure with pores smaller than the other two hydrogel samples. It seems that in the formulations the surface morphology was highly influenced by the crosslinker chemical structure"

Response: Line 361, According to the suggestion of the reviewer, the sentence was rephrased. Thank you very much for this observation.

Reviewer 2 Report

No further comments.

Author Response

N/A

Round 3

Reviewer 1 Report

I recommend its acceptance.